# Teacher attitudes toward inclusion of students with disabilities in Jeddah elementary schools

**Abdulaziz Alsolami** [1]*, **Michael Vaughan** [2]

1 Department of Special Education, King Abdulaziz University, Jeddah, Saudi Arabia, 2 Department of Social Work, Saint Louis University, Saint Louis, Missouri, The United States of America

☉ These authors contributed equally to this work.

* asalsolmi@kau.edu.sa

## Abstract

This study aims to investigate teachers' attitudes toward the inclusion of students with special educational needs (SEN) in general school settings in Jeddah, Kingdom of Saudi Arabia. Data were collected from a sample of teachers in Jeddah. A stratified random technique was utilized to choose the target sample, however, 187 teachers completely responded on the study questionnaire. Descriptive statistics were utilized to assess teachers' demographic profile. Also, Analysis of variance (ANOVA) was utilized to examine the association between teachers' demographic characteristics and their perceptions of SEN. Results revealed that teachers are more likely to believe that they lack the academic qualifications and training required for the inclusion of students with disabilities. They reported being trained only to teach students with learning difficulties rather than intellectual disabilities. The reported barriers to inclusion are lack of appropriate educational materials, limited time to devote sufficient attention to students with SEN, limited knowledge regarding SEN, and classrooms that are not designed for students with disabilities. The study findings therefore indicate the need for not only training teachers to teach students with disabilities but also, more importantly, ways to implement these strategies more effectively in typical school settings.

## Introduction

Inclusive education is extremely important for students with special educational needs (SEN) so that they can benefit from building both cognitive and non-cognitive skillsets supported in a formal learning environment [1]. Cognitive skills include the core skills of use, remember, learn, think, and pay attention to. Reasoning and non-cognitive skills include behavior and interaction with others while in school, which include communication, motivation, and interpersonal and social skills. Both cognitive and non-cognitive skills have been shown to be crucial to childhood development [1]. Students with special education needs (SEN) are those who suffer from any of the following: "specific impairments and disabilities listed in the law are mental retardation (also known as intellectual disabilities); hearing impairments, including deafness; speech or language impairments; visual impairments, including blindness; serious

**Data Availability Statement:** All relevant data are within the paper and its Supporting information files.

**Funding:** This work was supported by the Deanship of Scientific Research (DSR), King Abdulaziz University, Jeddah, under the grant number (D-711-324-1441). The authors, therefore, gratefully acknowledge the DSR for financial support. The funders had no role in study design, data collection and analysis, decision to publish, or preparation of the manuscript.

**Competing interests:** Enter: The authors have declared that no competing interests exist.

emotional disturbances; orthopedic impairments; autism; traumatic brain injury; other health impairments; specific learning disabilities; deaf-blindness; and multiple disabilities requiring special education and related services" [2].

Inclusive education has prevailed in Saudi Arabia for more than six decades. In 1947, the St. Vincent's home for Mentally Defective children made history as the first school to be approved by the state as a special school for children [3]. Prior to this, special schools had been established during the 16th and 19th centuries. When circumstances made it impossible to set up such special schools, special classes, particularly focusing on the education of pupils with SEN, were conducted. By the mid-1970s, a network of more than 100 special schools was established, and the number of special classes in mainstream schools grew significantly [4]. Since then, there has been a significant increase in the demand and need to include children with SEN in mainstream schools. Similarly, the Kingdom of Saudi Arabia used integration strategies wherein students with disabilities could be with their peers for a part of the day and study in a special class for the rest of the period in school [4]. In 2019, the Ministry of Education announced the implementation of inclusive education [5].

"There are strong empirical grounds for believing that teachers can and do make a difference and that consistent high quality teaching, supported by strategic professional development, can and does deliver dramatic improvements in student learning" [6]. Therefore, teachers must be sympathetic toward the students they teach. General education teachers play a significant role in improving the special education process. Teachers' beliefs and attitudes are tremendously important in their role of implanting the right attitude in each learner. Developing the right attitude helps students perform better in school, make independent decisions, and socially interact with others [7]. Likewise, individuals' attitudes can be recognized by their thoughts, feelings, and actions toward other persons or events they come across in their daily work. Previous studies have highlighted the ability of teachers to influence the whole education process via their attitudes [8].

Teachers' attitudes toward the inclusion of students with SEN in schools has drawn a range of mixed reactions. Although children with SEN face different educational problems, they share similar experiences in overcoming inclusion barriers within the context of the school and elsewhere [9]. Despite the call to include all students with SEN, teachers' attitude toward integrating students with SEN into the public education system remains a matter of serious concern.

This paper therefore aims to investigate the attitudes of general education teachers toward the inclusion of students with SEN in Jeddah, Saudi Arabia by outlining the existing barriers that impede the implementation of inclusive education. The study utilizes a descriptive and inferential research design to investigate and further shape the expected results of this study.

## Background

### Teachers' attitudes towards inclusion

Inclusive education is a "new approach towards educating children with disability and learning difficulties with normal children under the same roof. It brings all students together in one classroom and community, regardless of their strengths or weaknesses in any area, and seeks to maximize the potential of all students" [10]. Teachers' attitudes toward inclusive education have elicited intense debate among educators. For instance, a study [11] explored teachers' attitudes toward the inclusion of students with reference to their demographics (age and gender) and occupational stress levels. They collected data from a total of 208 secondary and primary school teachers who worked in urban and suburban areas and found that younger teachers demonstrated positive attitudes toward inclusive education when compared to their older colleagues. Furthermore, teachers' attitudes toward inclusive education were found to be partially

correlated to stress. For instance, teachers experiencing heightened levels of stress displayed a less positive attitude toward inclusion. Interestingly, gender was not found to significantly influence teachers' attitudes toward inclusive education.

A similar study [12] found that teachers' were supportive of inclusive education, but only partially considering the numerous challenges associated with its implantation. Other studies have also reported mixed attitudes of teachers toward inclusion [13, 14]. In a study exploring the association between background variables of teachers (gender, age, education) and their subsequent attitudes toward student inclusion, age was found to be a significant influencing factor with older teachers being less receptive [15]. In another study, gender was found to significantly influence attitudes toward inclusive education [16]. However, the authors highlight the need for further research to replicate the gender differences observed.

In addition to the influence of teachers' background characteristics, training also has been reported to play a key role in influencing teachers' attitudes towards inclusive education [17, 18]. For instance, teachers who received training on inclusive education were found to have a positive attitude toward inclusive education compared to those who did not [17, 18]. Therefore, training is viewed as fundamental for inculcating positive attitudes toward inclusive education. Although not without difficulties, prior research has highlighted that proper training provides teachers with the necessary knowledge to espouse school curricula in a manner that suits the needs of disabled children [19], for example, by making classes interactive and attractive through the use of colors, graphics, and other tools that establish higher levels of understanding among students. Moreover, teachers who receive training on inclusive education are also more equipped to manage the behavior of students. Although barriers exist, Olinger [20] finds that successful inclusion requires teamwork between special and general education providers. However, the same study showed that in the general environment, inclusion may not be suitable for all students in all circumstances.

## Importance of inclusion

The significance of including children with SEN in general settings cannot be underestimated. Thompson et al. [21] examined the need for including children with SEN in the general educational setting and reported that planning, arranging, and integrating feasible and effective support for students with various intellectual disabilities to access the general education setting and curriculum were key to ensuring high-quality education. Inclusive education is essential for maximizing students' performances in spite of their reported SEN [22]. When effectively implemented, inclusive education prepares children in a better way to face life through more appropriate learning opportunities. Contreras et al. [23] explored the effect of incorporating students with SEN on their peers' academic achievement. On average, the study found that the proper integration of students with SEN in a classroom setting improved their academic performance.

To determine the role and place of inclusive education in advancing the sustainable development of society, Fedulova et al. [24] conducted a comparative analysis in Russia of existing teaching practices for students with SEN. Despite the fundamental role of inclusive education in promoting sustainable development, the study reported low levels of readiness of the national governments as well as its educational institutions in implementing inclusive education for students with SEN. This indicates the persistence of developmental problems in governmental and educational institutions.

## Barriers to inclusion

Several studies highlight the presence of barriers that hinder inclusive education. One such barrier includes work-related stress, which has been shown to influence a teacher's perception

of inclusive education [25, 26]. Sukbunpant et al. [27] repeatedly cite stress as one of the deterrents or hindrances to a proper implementation of inclusive education. Extra workload and lack of cooperation with parents were cited as the primary sources of stress contributing to negative attitudes toward inclusive education [27]. However, Monsen et al. [28] found no significant association between the level of stress and attitudes toward inclusive education. Another factor that hinders inclusive education is the lack of training for teachers. When teachers did not receive SEN training, they generally displayed negative perceptions toward inclusion [25, 26]. Further, Contreras et al. [23] cited lack of resources as a major barrier. Reducing workplace stressors, providing SEN training to teachers, and provision of adequate resources in classroom settings are central to promoting inclusion.

## The research problem and its context

The Kingdom of Saudi Arabia has taken many initiatives to ensure that educational services are accessible to students with SEN. Although Saudi law guarantees free general education for all children irrespective of their backgrounds or learning abilities [29], analyzing the special education policy for inclusion of students with SEN revealed that the policy components do not support the inclusion of students with SEN in the right earnest. Students with SEN were considered different from their peers in the general setting [30]. A recent report issued by the Ministry of Education revealed an increase in the percentage of students with SEN and that only a limited number of schools accommodate such students [5]. The report estimated that the total number of students with SEN receiving education either at a private or public institution is approximately 70% [5]. Following the release of its report, the Ministry of Education highlighted the urgent need for establishing 100 schools to accommodate the remaining students with SEN in Saudi Arabia [5]. This highlights the importance of approaching elementary/primary school teachers to examine their attitudes toward inclusion. Based on the investigator's knowledge, no previous studies have discussed this topic in the context of Jeddah's Elementary Schools (JES).

## Purpose of the study

Elementary schools are considered the cornerstone for children to acquire fundamental knowledge. Providing students with such knowledge in early elementary school contributes positively to their success in subsequent stages of their education. The Ministry of Education, Saudi Arabia, considers elementary schools an essential educational phase where values and ethics are instilled in students [31]. This highlights the fundamental importance of teachers' attitudes toward the inclusion of students with SEN in the foundational phase of education and the need for further educational improvement. Since there is no literature on the implementation of inclusive education in JES, this study aims to investigate teachers' attitudes toward the inclusion of students with a disability in JES. This study is important because it supports the need for equality between students with disabilities and those without in the field of education in Saudi Arabia. This study offers decision-makers an opportunity to understand the status of inclusive education in JES. It also provides a baseline of information for future studies on inclusive education for students with disabilities in Saudi Arabia and add to the storehouse of cross-national comparative research on the topic. Given this backdrop, the following research questions drive this study. First, what are the current teacher attitudes toward the inclusion of learners with disabilities in general education schools? Second, does the type of school provision (full inclusion, partial inclusion, no provision for inclusion) influence teachers' attitudes toward inclusion? Third, what are the factors associated with teacher attitudes about the inclusion process? Finally, what are the teachers' perceptions on obstacles to inclusion and what do they consider are ideal methods to improve the inclusion process?

## Materials and methods

### Research design

For the current study, a descriptive and inferential research design was chosen to examine the attitude of general education teachers toward the inclusion of students with disabilities in JES. Descriptive research allows for quantitative research that involves a careful description of educational phenomena [32] and for its examination in an accurate manner [33]. Data were gathered from a survey administered beginning March 29, 2021.

Approval was obtained for the quantitative research from the Department of Special Education, King Abdulaziz University. To get the permission from the participants' job in this study, the Department of Special Education contacted the Board of Education in Jeddah where the study was conducted by distributing a survey link to teachers of special education in Jeddah. The participants of this study were special education teachers and a written consent was obtained when they at the beginning clicking the box asserting their consent of their participation in this study. All subjects completed the voluntary consent section in the questionnaire and were assured of confidentiality. The participants of the study were drawn from a total of 741 educators who taught students with SEN in Jeddah. A stratified random technique was utilized to draw the required sample size for this study. Out of the 253 participants who received the questionnaires, only 190 responses were recorded. After excluding the incomplete responses, the sample size was reduced to 178 participants representing about 70% of the target sample.

### Measurement

To collect the data required to achieve the research objectives, a survey questionnaire was designed and distributed among the teachers of students with SEN in Jeddah. The teachers (participants) in this study were randomly selected as mentioned earlier, and 178 completed surveys were collected.

According to Heale & Twycross [34], validity is the extent to which a concept is accurately measured in a quantitative study, whereas reliability is the extent to which an instrument displays the consistency of a measurement [35]. To ensure the efficacy of the instrument used to collect data for our study, the survey was pilot-tested among 30 participants to ensure its reliability and validity. To verify the reliability of the current study's instrument, Cronbach's alpha test was used to check for the internal consistency of the instrument. Table 1 indicates that the levels obtained, close to the value of 1 (0.917), are satisfactory in each dimension. Furthermore, the survey subscale also shows suitable reliability as the reliability statistics extended between 0.682 and 0.956.

Table 2 shows that all items included in the survey are significantly correlated with the related subscale, which indicates that all items achieved the measurement objective. Thus, all items in the survey were maintained.

**Table 1. Instrument reliability analysis.**

| Subscales | Number of Items | Cronbach's Alpha Coefficient |
|---|---|---|
| The educator's attitudes toward inclusion | 7 | 0.682 |
| Teacher's preparedness for teaching | 12 | 0.835 |
| Best situation or environment for teaching students with SEN | 7 | 0.815 |
| Barriers for inclusion | 20 | 0.956 |
| Overall reliability | 46 | 0.917 |

**Table 2. Validity of the instrument.**

| Item No. | Correlation coefficient | Item No. | Correlation coefficient | Item No. | Correlation coefficient | Item No. | Correlation coefficient |
|---|---|---|---|---|---|---|---|
| 1 | 0.750** | 13 | 0.431* | 25 | 0.700** | 37 | 0.762** |
| 2 | 0.493** | 14 | 0.717** | 26 | 0.673** | 38 | 0.869** |
| 3 | 0.645** | 15 | 0.769** | 27 | 0.694** | 39 | 0.732** |
| 4 | 0.585** | 16 | 0.485** | 28 | 0.662** | 40 | 0.824** |
| 5 | 0.361* | 17 | 0.617** | 29 | 0.719** | 41 | 0.778** |
| 6 | 0.676** | 18 | 0.793** | 30 | 0.683** | 42 | 0.729** |
| 7 | 0.621** | 19 | 0.758** | 31 | 0.700** | 43 | 0.792** |
| 8 | 0.628* | 20 | 0.684** | 32 | 0.650** | 44 | 0.723** |
| 9 | 0.532** | 21 | 0.678** | 33 | 0.663** | 45 | 0.863** |
| 10 | 0.706** | 22 | 0.677** | 34 | 0.849** | 46 | 0.734** |
| 11 | 0.778* | 23 | 0.744** | 35 | 0.613** | | |
| 12 | 0.366* | 24 | 0.683** | 36 | 0.769** | | |

** indicates that the correlation coefficient is significant at the 0.01-level;

* indicates that the correlation coefficient is significant at the 0.05-level.

## Data analysis

Of 190 questionnaires, only 178 responses were analyzed. Incomplete responses (12) were excluded. Once the data were collected, SPSS Version 20 was used for data analysis. Descriptive and inferential statistical methods were used. Descriptive statistics assessed the frequencies, percentages, means, and standard deviations of all the variables [36]. Analysis of variance (ANOVA) was conducted as an inferential statistical method to examine the association between teachers' demographic characteristics and their perceptions of SEN.

**Analysis of demographic characteristics.** A total of 178 teachers participated in this study. Among the participants, 58.4% were older than 40 years, nearly 30% were aged between 30 and 40 years, and less than 15% were aged below 30 years. According to Table 3, teachers with over 16 years of experience constituted more than half (59.5%) of the participants. Further, 43.9% worked at ordinary schools providing private education classes and 40.4% worked in ordinary schools that did not provide private education classes. Only 15.7% worked in integrated schools. Most teachers (66.3%) had no academic qualifications in special education, 21.9% had bachelor degrees, 7.3% had master's degrees, and only 4.5% had a diploma. Almost 58% hadn't received any training on special education. Nearly 67% did not have any contact with students with SEN outside school.

## Results

### What were the educators' attitudes toward the inclusion of learners with disabilities in the general education schools?

To examine the educators' attitudes toward the inclusion of learners with disabilities in general education schools, participants were asked to respond according to their degree of agreement (strongly disagree = 1 to strongly agree = 5). The overall mean score results in Table 4 show that the teachers agreed with items whose ranks fell between 1 and 5 in the mean range of 2.48–3.24 and disagreed on items with ranks that fell between 6 and 11 in the mean range of 2.04–2.42. The overall mean was less than the cut-off mean (i.e., 2.47< 2.50), which indicates that most teachers had negative attitudes towards the inclusion of students with SEN in regular classrooms.

**Table 3. Teachers' demographic profile.**

| | Frequency | Percent |
|---|---|---|
| *Age (years)* | | |
| 21–25 | 3 | 1.7 |
| 26–30 | 17 | 9.6 |
| 31–40 | 54 | 30.3 |
| More than 40 | 104 | 58.4 |
| *Experience in years* | | |
| Less than two years | 7 | 3.9 |
| 2–5 years | 11 | 6.2 |
| 6–10 years | 30 | 16.9 |
| 11–15 years | 24 | 13.5 |
| 16–20 years | 33 | 18.5 |
| More than 20 years | 73 | 41.0 |
| *School type* | | |
| Integrated school | 28 | 15.7 |
| Ordinary school with private education classes | 78 | 43.8 |
| Ordinary school without private education classes | 72 | 40.4 |
| *Academic qualification in special education* | | |
| None | 118 | 66.3 |
| High School Diploma | 8 | 4.5 |
| Bachelor's degree | 39 | 21.9 |
| Master's degree | 13 | 7.3 |
| *Having training courses on special education* | | |
| No | 103 | 57.9 |
| Yes | 75 | 42.1 |
| *Having contact time with students with SEN other than work time* | | |
| No | 118 | 66.3 |
| Yes | 60 | 33.7 |

To determine whether teachers' perspectives toward the inclusion of students with SEN were related to age, experience, and school type, an ANOVA test was conducted. The results in Table 5 revealed that there was no statistical significance between the teachers' perceptions and their age, experience levels, or the type of school they worked in.

## Influence of the type of school provision on the teachers' attitudes toward inclusion

A one-way ANOVA between subjects was conducted to examine the effect of the type of school provision (full or partial inclusion, or no provision for inclusion) on teachers' attitudes toward inclusion. The effect of the independent variable on the dependent one was not found statistically significant at the $p < 0.05$ level. The results in Table 6 show that all values of F-Statistics are not statistically significant as all P-values are greater than the significant level ($\alpha = 0.05$).

## Educators' perceptions on barriers to inclusion and methods to improve the inclusion process

To examine the educators' perceptions on barriers to inclusion, participants were asked to specify answers according to their degree of agreement with the proposed items on the

**Table 4. Summary of teachers' attitudes toward inclusion of students with special educational needs.**

| Attitudes | Mean | SD | Ranking |
|---|---|---|---|
| *Students with special educational needs are best served through special separate classes* | 2.42 | 1.26 | 6 |
| *The challenges of being in an ordinary classroom can promote the academic growth of a child with special educational needs* | 2.30 | 1.02 | 9 |
| *Inclusion offers mixed-group interactions that will foster the understanding and acceptance of differences* | 2.12 | 0.99 | 11 |
| *Isolation in a special class has a negative effect on the social and emotional development of a student with special educational needs* | 2.48 | 1.16 | 5 |
| *A child with special educational needs will probably develop academic skills more rapidly in a special classroom than in an ordinary classroom* | 2.35 | 1.05 | 8 |
| *Contact between regular students and students with special needs in the classroom may be harmful* | 2.72 | 1.17 | 3 |
| *Including a child with special educational needs will promote his/her social independence* | 2.39 | 1.05 | 7 |
| *The inclusion of students with special educational needs can benefit regular students* | 2.65 | 1.14 | 4 |
| *Inclusion is likely to have a negative effect on the emotional development of a child with special educational needs* | 2.77 | 1.08 | 2 |
| *A child with special educational needs will be socially isolated by other students* | 3.24 | 1.00 | 1 |
| *Students with special educational needs should be given every opportunity to participate in an ordinary classroom setting, wherever possible* | 2.04 | 1.06 | 12 |
| *The presence of students with special educational needs can promote the acceptance of difference among other students* | 2.15 | 0.94 | 10 |
| Overall mean value | 2.47 | 0.52 | |

**Table 5. Perceived barriers to the implementation of inclusion.**

| No. | Barriers | Mean | SD | Ranking |
|---|---|---|---|---|
| 1 | Inadequate pre-service preparation of teachers | 3.58 | 1.32 | 14 |
| 2 | Overload on teachers | 3.83 | 1.26 | 10 |
| 3 | Classrooms do not accommodate students with disabilities | 4.10 | 1.17 | 4 |
| 4 | Absence of regulations that support inclusion | 3.72 | 1.17 | 12 |
| 5 | Teachers' negative attitudes | 3.39 | 1.12 | 16 |
| 6 | Resistance among administrators | 3.28 | 1.09 | 18 |
| 7 | Non-acceptance by other parents | 3.19 | 1.19 | 19 |
| 8 | Little knowledge of special educational needs | 4.11 | 1.00 | 3 |
| 9 | Lack of experience vis-à-vis inclusion | 4.01 | 1.02 | 6 |
| 10 | Class size or large teacher/pupil ratio | 4.05 | 1.22 | 5 |
| 11 | Limited time for teachers to give sufficient attention to students with SEN | 4.12 | 1.10 | 2 |
| 12 | Lack of equipment and appropriate educational materials | 4.33 | 0.90 | 1 |
| 13 | Non-acceptance by parents of SEN students | 3.17 | 1.21 | 20 |
| 14 | Behavior management | 3.90 | 1.03 | 8 |
| 15 | Rigidity in curriculum design and examination | 3.69 | 1.05 | 13 |
| 16 | Lack of concern for diversity of interests and abilities | 3.49 | 1.11 | 15 |
| 17 | Inadequate in-service training for teachers | 3.91 | 1.03 | 7 |
| 18 | Non-acceptance by other students | 3.34 | 1.08 | 17 |
| 19 | Absence of an educational policy for inclusion in Saudi Arabia and absence of a clear vision for change | 3.88 | 0.98 | 9 |
| 20 | Inadequate funding | 3.79 | 1.20 | 11 |
| | **Overall mean value** | **3.74** | **0.68** | |

**Table 6. ANOVA results concerning teachers' perspectives on inclusion related to teachers' ages, experience by years, and school types.**

| | Educators' attitudes toward inclusion M(SD) | Teachers' preparedness for teaching M(SD) | Barriers to the implementation of inclusion M(SD) |
|---|---|---|---|
| *Age of respondents* | | | |
| 21–25 years | 2.36(0.63) | 2.38(0.59) | 4.25(0.53) |
| 26–30 years | 2.54(0.48) | 2.01(0.70) | 3.82(0.51) |
| 31–40 years | 2.45(0.55) | 2.09(0.68) | 3.76(0.67) |
| >40 years | 2.47(0.50) | 2.10(0.64) | 3.71(0.72) |
| *Respondents' experience in years* | | | |
| 2–5 years | 2.42(0.53) | 1.96(0.67) | 3.88(0.59) |
| 6–10 years | 2.44(0.49) | 2.07(0.63) | 3.74(0.70) |
| 11–15 years | 2.51(0.60) | 2.04(0.75) | 3.66(0.49) |
| 16–20 years | 2.50(0.53) | 2.36(0.72) | 3.54(0.85) |
| >20 years | 2.46(0.49) | 2.04(0.57) | 3.83(0.66) |
| *School Type* | | | |
| Integrated School | 2.51(0.60) | 2.08(0.79) | 3.51(0.76) |
| Ordinary school with private educational classes | 2.54(0.47) | 2.15(0.63) | 3.77(0.61) |
| Ordinary school without private educational classes | 2.37(0.51) | 2.04(0.62) | 3.80(0.72) |

questionnaire (strongly disagree = 1 to strongly agree = 5). Table 7 shows that the overall mean score is greater than the cut-off mean (i.e., 3.74>2.50). Thus, most teachers agreed with the various barriers indicated in the table. Among the major barriers found were the lack of equipment and appropriate educational materials (mean = 4.33, SD = 0.90). Thus, almost all

**Table 7. One-way analysis of variance (ANOVA) to examine the influence of school types on teachers' attitudes towards inclusion of students with SEN.**

| | School types | | | | | |
|---|---|---|---|---|---|---|
| | Integrate school | | Ordinary school with private education classes | | Ordinary school without private education classes | |
| **Attitudes** | **M** | **SD** | **M** | **SD** | **M** | **SD** |
| *Students with special educational needs are best served through special separate classes* | 2.39 | 1.31 | 2.60 | 1.30 | 2.22 | 1.17 |
| *The challenges of being in an ordinary classroom can promote the academic growth of a child with special educational needs* | 2.32 | 1.16 | 2.29 | 1.01 | 2.29 | 0.99 |
| *Inclusion offers mixed-group interactions that will foster the understanding and acceptance of differences* | 2.29 | 1.27 | 2.10 | 0.97 | 2.07 | 0.89 |
| *Isolation in a special class has a negative effect on the social and emotional development of a student with special educational needs* | 2.61 | 1.26 | 2.51 | 1.11 | 2.39 | 1.17 |
| *A child with special educational needs will probably develop academic skills more rapidly in a special classroom than in an ordinary classroom* | 2.25 | 1.24 | 2.53 | 1.02 | 2.19 | 1.00 |
| *Contact between regular students and students with special needs in the classroom may be harmful* | 2.50 | 1.37 | 2.94 | 1.14 | 2.58 | 1.08 |
| *Including a child with special educational needs will promote his/her social independence* | 2.46 | 1.29 | 2.45 | 0.99 | 2.29 | 1.03 |
| *The inclusion of students with special educational needs can benefit regular students* | 2.82 | 1.31 | 2.72 | 1.13 | 2.50 | 1.07 |
| *Inclusion is likely to have a negative effect on the emotional development of a child with special educational needs* | 2.75 | 1.35 | 2.94 | 1.07 | 2.60 | 0.96 |
| *A child with special educational needs will be socially isolated by other students* | 3.21 | 1.13 | 3.33 | 0.91 | 3.14 | 1.05 |
| *Students with special educational needs should be given every opportunity to participate in an ordinary classroom setting, wherever possible* | 2.21 | 1.23 | 1.96 | 1.06 | 2.06 | 1.01 |
| *The presence of students with special educational needs can promote the acceptance of difference among other students* | 2.36 | 1.13 | 2.13 | 0.86 | 2.10 | 0.95 |
| **Overall mean value** | 2.51 | 0.60 | 2.54 | 0.47 | 2.37 | 0.51 |

teachers strongly agreed that both factors were strong barriers to the successful implementation of including students with special educational needs.

Based on their rankings, other key barriers that most teachers identified included limited time for teachers to pay sufficient attention to students with SEN (mean = 4.12, SD = 1.10), teachers' limited knowledge of SEN (mean = 4.11, SD = 1.0), classrooms not being equipped to accommodate students with disabilities (mean = 4.10, SD = 1.17), and having a large number of pupils in class (mean = 4.05, SD = 1.22).

## Discussion

Given their central role in the education process, it is important to understand teacher attitudes toward the inclusion of students with disabilities, especially among students in non-western contexts where less research has been conducted. This study sought to investigate teachers' attitudes toward the inclusion of elementary/primary school students and to uncover the main barriers and obstacles to the inclusion of students with special educational needs, which can provide key information to educational policymakers in Saudi Arabia.

This study found that the major barriers for inclusion are the lack of equipment and appropriate educational materials, and the availability of limited time for teachers to pay sufficient attention to students with SEN. Consistent with prior research [12], such barriers may explain why some teachers only partially support inclusive practices. Also, insufficient time to pay attention to SEN can be a disruptive factor to the quality teaching provided to them [28]. In addition to the above reason, teachers cite the following barriers including lack of textbooks, teaching aids and training equipment [28]. There was little knowledge of special educational needs (e.g., lack of in-service training). For example, the results revealed that most teachers (approximately two-thirds) had no academic qualifications in special education, and most of them (58%) had no training at all to teach students with special needs. Most teachers reported that their work schedule was demanding and full and as a result, they had little time to devote to students with SEN. Classrooms were not very accommodative for students with disabilities. Most teachers confirmed the lack of funding and policy and accurate training programs as major factors or barriers that prevented the successful implementation of inclusion. Various studies that have examined the main barriers to the implementation of inclusion have tended to confirm the lack of training as a salient factor along with necessary resources to implement such training into action [23, 25, 26]. Training enhances teacher effectiveness and reduces negative perceptions that teachers may have toward inclusion. Finally, it is important to build the course structure on the students' evaluations of teaching and recommendation to improve the skills necessary for their potential career [37].

## The theoretical contribution and practical implication

The current study has both a theoretical contribution and practical implication. The theoretical contribution highlights the inclusive learning's role to minimize the least restrictive environments for students with disabilities compared to their peers. Identifying the teachers' attitudes of inclusive education can help promote the engagement of students with disabilities with general education classrooms. The study practical implications for policy and practice. First, to teach students with special educational needs in primary school settings successfully, it is important to empower teachers through comprehensive training. Opening up new opportunities for teachers to qualify academically in teaching students with special educational needs is important for successful inclusion programs. This may be implemented efficiently through the "each one teach one" strategy whereby a lead teacher undergoes these qualifications and passes on the training to other teachers and staff within the school. Providing

teachers with sufficient knowledge with respect to inclusion is important. Moreover, engaging a new model of education concentrating on interactivity and active collaborative learning can have a positive influence on students' critical thinking [7]. Therefore, Schools should also invest in the equipment and educational materials that can support students with special educational needs. The latter may require advocacy on part of the teachers and some administrative creativity. Schools and educational systems should commit to this ideal.

## Limitations

Findings from this study should be interpreted in light of several limitations. First, all data were derived from self-reported information and are thus susceptible to recall or other biases. Second, although the data are unique, their cross-sectional structure precludes any causal determination vis-à-vis the relationship between attitudes and inclusion practice. Longitudinal investigations should be conducted to disentangle the temporal ordering needed to identify the attitudes that affect inclusionary practices. Finally, the results presented here pertain to the youth in one city and may not be representative of teachers across the nation. Future research would benefit from greater depth in assessing students, teachers, administrators, and parents' attitudes and triangulate these results toward creating a complete picture of inclusionary attitudes.

## Conclusion and future work

Building inclusive classrooms for children with disabilities in ordinary school settings is challenging for teachers. Our study revealed that the type of school has no significant influence on teachers' attitudes towards inclusion. It found that, among the barriers that impede the implementation of inclusion of students with special educational needs in primary school settings, the lack of equipment and appropriate educational materials, availability of limited time for teachers to give sufficient attention to students with SEN, poor knowledge of special educational needs, and classrooms that cannot accommodate students with disabilities are significant. Class size or large teacher/pupil ratio, lack of experience regarding inclusion, and inadequate in-service training for teachers were also barriers. These findings align with results on teacher attitudes in western contexts, suggesting that there may be some degree of universality among teachers while confronting inclusive education. Given that inclusive education maximizes performance and better prepares students for life [21], identifying efficient and effective ways to improve teacher capabilities is a pressing educational policy matter. Future work should focus on providing appropriate education for our students regardless of their disability, race, ethnicity, and color require positive societal attitude associated with united collaborative efforts among all stakeholders such as parents, schools, and the legislature. If such effort tailed with positive attitude and suitable environment that will embrace our students to achieve ultimate educational goal which align with the aim of inclusive education to avoid the exclusion from the general education system [38]. Future work should also focus on conducting quasi-experimental studies to assess the outcomes for children with disabilities among schools that have these resources and teacher training protocols, and to compare these schools with those that are yet to implement these changes. Although not as strong of a design as true experimentation, this approach is practical, feasible, and will provide an initial causal inferences.

## Supporting information

**S1 Data.**
(XLSX)

## Author Contributions

**Conceptualization:** Abdulaziz Alsolami.

**Formal analysis:** Michael Vaughan.

**Investigation:** Abdulaziz Alsolami, Michael Vaughan.

**Methodology:** Michael Vaughan.

**Resources:** Abdulaziz Alsolami.

**Software:** Abdulaziz Alsolami.

**Supervision:** Abdulaziz Alsolami.

**Validation:** Abdulaziz Alsolami.

**Writing – original draft:** Michael Vaughan.

**Writing – review & editing:** Abdulaziz Alsolami, Michael Vaughan.

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
