## [Decision Letter · Decision Letter 0]

10 May 2022

PONE-D-21-41070Teacher Attitudes Toward Inclusion of Students with Disabilities in Jeddah Elementary SchoolsPLOS ONE

Dear Dr. Alsolami,

Thank you for submitting your manuscript to PLOS ONE. After careful consideration, we feel that it has merit but does not fully meet PLOS ONE’s publication criteria as it currently stands. Therefore, we invite you to submit a revised version of the manuscript that addresses the points raised during the review process.

We look forward to receiving your revised manuscript.

Kind regards,

Ender Senel, PhD

Academic Editor

PLOS ONE

Journal Requirements:

3. Please ensure that you have specified (1) whether consent was informed, (2) what type you obtained (for instance, written or verbal, and if verbal, how it was documented and witnessed). If your study included minors, state whether you obtained consent from parents or guardians. If the need for consent was waived by the ethics committee and (3) If you are reporting a retrospective study of medical records or archived samples, please ensure that you have discussed whether all data were fully anonymized before you accessed them and/or whether the IRB or ethics committee waived the requirement for informed consent. If patients provided informed written consent to have data from their medical records used in research, please include this information.

This work was supported by the Deanship of Scientific Research (DSR), King Abdulaziz University, Jeddah, under the grant number (D-711-324-1441). The authors, therefore, gratefully acknowledge the DSR for financial support. 

 This work was supported by the Deanship of Scientific Research (DSR), King Abdulaziz University, Jeddah, under the grant number (D-711-324-1441). The authors, therefore, gratefully acknowledge the DSR for financial support. 

This work was supported by the Deanship of Scientific Research (DSR), King Abdulaziz University, Jeddah, under the grant number (D-711-324-1441). The authors, therefore, gratefully acknowledge the DSR for financial support. 

7. Please amend either the abstract on the online submission form (via Edit Submission) or the abstract in the manuscript so that they are identical.

8. Please include your full ethics statement in the ‘Methods’ section of your manuscript file. In your statement, please include the full name of the IRB or ethics committee who approved or waived your study, as well as whether or not you obtained informed written or verbal consent. If consent was waived for your study, please include this information in your statement as well. 

Reviewers' comments:

Reviewer's Responses to Questions

**Comments to the Author**

1. Is the manuscript technically sound, and do the data support the conclusions?

Reviewer #1: Yes

Reviewer #2: Yes

2. Has the statistical analysis been performed appropriately and rigorously? 

Reviewer #1: Yes

Reviewer #2: Yes

3. Have the authors made all data underlying the findings in their manuscript fully available?

Reviewer #1: Yes

Reviewer #2: Yes

4. Is the manuscript presented in an intelligible fashion and written in standard English?

Reviewer #1: Yes

Reviewer #2: No

5. Review Comments to the Author

Reviewer #1: I enjoyed reading your paper. It is clear and concise. I also learned much about special education in Saudi Arabia, an area of the world where I know little.

One minor correction I observed is below:

Line 61: should be General education instead of General/ education

Reviewer #2: I am pleased to have had the opportunity to review this interesting study.

The topic of your study is important, current, and relevant, as special education is by-and-large a neglected area in education worldwide, resulting in direct and sometimes serious effects on a great number of students, their families, and the population in general. However, in order for the current manuscript to be considered for publication in PLOS ONE, extensive revisions would need to be made.

First and foremost, a strong case needs to be made at the start of the paper for WHY a study such as this is relevant and generally applicable outside of Jeddah. How does it fit into the body of research on special education, and why did it need to be done? How will it help to move the field of research on special education forward? What is the state of research on special education worldwide, and why are further studies such as this one warranted? Have most of the studies thus far been done in the West? Further comments on this topic are included in the line-by-line commentary below.

Further, this is a study of students with special education needs, yet there is no description anywhere in the paper of exactly what constitutes a student with special education needs. Are they students with intellectual disabilities, learning difficulties (such as perhaps dyslexia), physical disabilities (blind, deaf, cerebral palsy, etc)? This information is fundamentally important to the study and needs to be clearly spelled out at the very start.

Additionally, this paper should be reviewed by a fluent English-speaker so that the language is clear. In parts of the manuscript as it is written it is difficult to understand exactly what the authors are trying to communicate. I would also suggest that a person fluent in English who is also quite familiar with this field look over the manuscript to be sure that languaging around the topic is current and sensitive to the population being studied (I do not have have the expertise for that, so the verbiage used in my suggestions may not be correct for example). Finally, there are some areas where there are significant logical leaps in parts of the paper that need to be addressed.

The conclusions could be built out more, which would also strengthen the case for why this study is important and relevant by expanding the context. For example, you can specify and contextualize societal issues created by lack of attention to special educational needs. Who does the burden of education fall on when schools fail to provide for students with special education needs? What are the implications of the fact that families (for example) must do this rather than schools? Etc. There could also be discussion of the reasons why communities or countries are failing to meet these needs. Is it cultural attitude? Lack of funding? Are the reasons different in different cultural contexts? How do the issues in Jeddah differ from those in other parts of the world? Etc. What are potential solutions to these issues that ought to be the subject of future research?

Line-by-line comments follow:

Abstract-49 Characteristics of “disability” need to be named before you do anything else. How is SEN defined? Give references to support your definition.

50 “it provides them with their learning demands”? Instead for example something like: … so that they can also benefit from building both cognitive and non-cognitive skills sets that a formal learning environment supports [reference]. Cognitive skills include ____ and non-cognitive skills include____. Both are shown to be crucial to childhood development [ref], and studies show that formal education is a key component to gaining those skills [reference].

51-59 Extremely confusing. You say that these schools have existed for 6 decades, then talk about the 16th and 19th centuries, which are clearly more than 6 decades ago! This section is also given no geographic or cultural context. Where were these schools first established? What was the impetus behind their establishment? Where and when did this concept spread to other parts of the world? How does Saudi Arabia fit into this narrative?

60-61 The first 2 sentences of the 2nd paragraph are stated as opinions. Re-state in a fact-based way, accompanied by references. For example something like: Prior studies indicate that a teacher’s dedication to their students as measured by _____ and ____ plays a significant role in the [social/emotional growth? Success?] of students as measured by _____ [refs]

69 Needs references

70-71 Needs references

72-73 Needs references

74-83 Your study is not a literature review. The section you have called “Literature Review” should be called “Background” – while it does consist of an overview of the literature in the research area, this is done to give context to your current study. Remove the final sentence of the paragraph that begins on line 70 (lines 74-77).

74-77 Replace this sentence with something that gives context to the importance of examining these things in Jeddah, Saudi Arabia. For example something like: As the second-largest city in Saudi Arabia, Jeddah is representative of the urban attitudes of Saudi educators towards special education. The present study aims to assess the attitudes of teachers regarding SEN in Jeddah in the [ ____ years? decades?] since the first special education [classroom? School?] was established.

85-91 You only need to list your reference here once.

85 Please define what is meant by “inclusive education”. Does this mean special needs students are included in a classroom together with regular students, or does it mean whole classrooms in a standard school that are dedicated to special education, or does it mean entire schools specifically dedicated to special education? Or some combination of these? Reading further in the paper makes it appear as though you mean including students with special education needs together in the same classroom with students who don’t have these needs. This needs to be defined and made clear the first time inclusion is mentioned in the paper (line 76?)

86 Where was this study conducted? Cultural context is needed. Was it in Saudi Arabia?

107 Omit “presumably due to status quo” or change to “which the authors attributed to a desire to maintain the status quo” if that was specifically hypothesized by those authors in that paper. The way you have it written it reads as your personal opinion rather than something based on a prior study.

96-106 Where were these studies conducted?

111-112 Where were these conducted?

130 Define “ID”

132 “performance” is very general here. How are you defining and evaluating these students’ performance?

139 What is meant by “society’s sustainable development”? Possibly this is a mistranslation, I cannot figure out what you mean by this.

141 see comments for line 139

138-144 Needs to be re-written as the point being made is not at all clear due to language and translation. Also was this study focused on Jeddah, or elsewhere? That needs to be specified as you go on to talk about governmental readiness and we need to know what government you refer to.

144 I don’t think you mean “society at-large” as I think you are referring specifically to governmental and educational institutions only.

187 Define JES

180-200 This is very good – here you have finally included a context and rationale for your study in Jeddah. This context however must be presented much earlier in the paper, as has been noted in prior comments above.

209 What year?

219-221 This sentence is not needed – the reader should already know validity and reliability measure. Also, this sentence mentions “quantitative study”, when you have already stated that your study is qualitative rather than quantitative

215-216 Before anything else you need to give some context to the development of the survey questions themselves. Where did they come from? Are they standard? (if so, provide a reference). Were they taken as an aggregate of those included on other similar studies? Did you come up with them on your own, and if so what was the rationale behind the choice of questions?

250-251 Please clarify: what are “ordinary schools” and what are “private education classes”? The table on p. 14 needs the same clarification.

252-254 Please describe what each of these education levels means. What level of education is required in order to be a teacher? 12 years of primary school? 12 years of primary school plus a bachelor’s degree? Etc. and when you say the majority of teachers “had none”, does this mean the teachers finished 12 years of primary education only, or does it mean something else? When you say 4.5% had “a diploma”, how is this different from the bachelor’s degree? The table on p. 14 needs the same clarification.

Table 4, p.15 Many of the questions listed in this table are not address or discussed in the paper. For example there is no discussion about teacher attitudes regarding the type of teaching environment most suited for SEN students (special separate classes, ordinary classroom, etc).

Again, I was pleased to have the opportunity to review this work, and I hope to have the opportunity to review revisions.

6. PLOS authors have the option to publish the peer review history of their article (what does this mean?). If published, this will include your full peer review and any attached files.

Reviewer #1: **Yes: **Jennifer Faux-Campbell

Reviewer #2: **Yes: **ML Mills

---

## [Author Response · Author response to Decision Letter 0]

10 Aug 2022

I have learned closely how to understand my paper, I followed exactly what you meant in your comments and that was great added knowledge to my paper. Thank you so much again.

---

## [Decision Letter · Decision Letter 1]

26 Sep 2022

PONE-D-21-41070R1Teacher Attitudes Toward Inclusion of Students with Disabilities in Jeddah Elementary SchoolsPLOS ONE

Dear Dr. Alsolami,

Thank you for submitting your manuscript to PLOS ONE. After careful consideration, we feel that it has merit but does not fully meet PLOS ONE’s publication criteria as it currently stands. Therefore, we invite you to submit a revised version of the manuscript that addresses the points raised during the review process.

ACADEMIC EDITOR: You are suggested to address the comments of reviewer 4.

We look forward to receiving your revised manuscript.

Kind regards,

Rano Mal Piryani, MBBS, MCPS, DTCD, MD, Fellowship in Med Education

Academic Editor

PLOS ONE

Journal Requirements:

Additional Editor Comments (if provided):

Authors has to address the comments of reviewer 4.

Reviewers' comments:

Reviewer's Responses to Questions

**Comments to the Author**

1. If the authors have adequately addressed your comments raised in a previous round of review and you feel that this manuscript is now acceptable for publication, you may indicate that here to bypass the “Comments to the Author” section, enter your conflict of interest statement in the “Confidential to Editor” section, and submit your "Accept" recommendation.

Reviewer #1: All comments have been addressed

Reviewer #3: (No Response)

Reviewer #4: (No Response)

2. Is the manuscript technically sound, and do the data support the conclusions?

Reviewer #1: Yes

Reviewer #3: No

Reviewer #4: Yes

3. Has the statistical analysis been performed appropriately and rigorously? 

Reviewer #1: Yes

Reviewer #3: No

Reviewer #4: Yes

4. Have the authors made all data underlying the findings in their manuscript fully available?

Reviewer #1: Yes

Reviewer #3: No

Reviewer #4: Yes

5. Is the manuscript presented in an intelligible fashion and written in standard English?

Reviewer #1: Yes

Reviewer #3: Yes

Reviewer #4: Yes

6. Review Comments to the Author

Reviewer #1: The paper is well written and clearly presents all collected data in an effective manner. Also, given my experience with inclusion in education, it is also an interesting read.

Reviewer #3: Page 18.Line 225. Table 2 shows that all items included in the survey are significantly correlated with the related subscale

The Correlation table is for two items e.g., the weight and height, or study hours and GPAs. the shown table is WRONG.

No 1 item has a correlation with other items??

page 18 Line 233 -234

Analysis of variance (ANOVA) was conducted as an inferential statistical method to examine the association between teachers’ demographic characteristics and their perceptions of SEN.

Line 240-241

Further, 43.9% worked at ordinary schools providing private and 40.4% worked in ordinary ... Only 15.7% worked in integrated schools. Comment: if 15% of teachers have been working in special schools, then results do not show true perceptions later authors mentioned Nearly 67% did not have any contact with students with SEN outside school.

Line 265. A one-way ANOVA between subjects was conducted to examine the effect of the type of 264 school provision on teachers’ attitudes toward inclusion. The effect of the independent variable on the dependent one was not found statistically significant at the p < 0.05 level.

Comment: ANOVA measures the mean differences of three or more groups. The given table does not reflect the ANOVA values. The effect is measured by Regression analysis. the said table is wrong

Table 6: One-Way analysis of variance (ANOVA) to examine the influence of school types on teachers' attitudes towards the inclusion of students with SEN.

Comment: The table must show the F value, df, P value, and Postdoc if there was a difference in means between the three groups (schools). The given Table 6 is wrong

Table 7 shows Ranking, usually, it is used when data is not normally distributed and a non-parametric test is applied. SPSS itself gives ranking

Overall, Inappropriate methodology, lack of content validity of study tool, and wrong statistical analysis make this study to be rejected

A good option is to use the qualitative method.

Reviewer #4: The paper “Teacher attitudes toward inclusion of students with disabilities in Jeddah elementary schools” is within the scope of the journal. I recommend revisions to the author and also request them to consider t.

1. In Abstract section, authors are requested to add some details about method (sample, data analysis tools and techniques used for this study).

2. In discussion section required more citations in term of linking their results with the support of state-of-the-art literature.

3. Authors are also requested to create new heading under discussion section i.e., “Implication of this study” and explain thoroughly about theoretical contribution and practical implication of this work.

4. Under Conclusion section, author create new sub-heading i.e., “Future work” and explain how other researcher can further extend this work in the future.

5. To strengthen this work, I suggest some articles to read and cite in your manuscript.

Abbas, A., Haruna, H., Arrona-Palacios, A., Camacho-Zuñiga, C., Núñez-Daruich, S., Enríquez de la O, J. F., … Hosseini, S. (2022). Students’ evaluations of teachers and recommendation based on course structure or teaching approaches: An empirical study based on the institutional dataset of student opinion survey. Education and Information Technologies. doi:10.1007/s10639-022-11119-z

Gonzalez-Cacho, T., & Abbas, A. (2022). Impact of interactivity and active collaborative learning on students’ critical thinking in higher education. IEEE Revista Iberoamericana de Tecnologias Del Aprendizaje, 17(3), 254–261. doi:10.1109/rita.2022.3191286

7. PLOS authors have the option to publish the peer review history of their article (what does this mean?). If published, this will include your full peer review and any attached files.

Reviewer #1: **Yes: **Jennifer Faux-Campbell

Reviewer #3: No

Reviewer #4: No

---

## [Author Response · Author response to Decision Letter 1]

20 Nov 2022

Please see the attached document.

---

## [Editor Report · Decision Letter 2]

1 Dec 2022

Teacher Attitudes Toward Inclusion of Students with Disabilities in Jeddah Elementary Schools

PONE-D-21-41070R2

Dear Dr. Alsolami,

We’re pleased to inform you that your manuscript has been judged scientifically suitable for publication and will be formally accepted for publication once it meets all outstanding technical requirements.

Kind regards,

Rano Mal Piryani, MBBS, MCPS, DTCD, MD, Fellowship in Med Education

Academic Editor

PLOS ONE
---

## [Editor Report · Acceptance letter]

22 Dec 2022

PONE-D-21-41070R2 

*Teacher attitudes toward inclusion of students with disabilities in Jeddah elementary schools*

Dear Dr. Alsolami:

I'm pleased to inform you that your manuscript has been deemed suitable for publication in PLOS ONE. Congratulations! Your manuscript is now with our production department. 

Kind regards, 

on behalf of

Dr. Rano Mal Piryani 

Academic Editor

PLOS ONE